# The Game of Tubulins

**DOI:** 10.3390/cells10040745

**Published:** 2021-03-28

**Authors:** Maria Alvarado Kristensson

**Affiliations:** Molecular Pathology, Department of Translational Medicine, Lund University, Skåne University Hospital, 20502 Malmö, Sweden; maria.alvarado-kristensson@med.lu.se; Tel.: +46-4033-7063

**Keywords:** GTPase, tubulins, γ-tubulin, meshwork, cytoskeleton

## Abstract

Members of the tubulin superfamily are GTPases; the activities of GTPases are necessary for life. The members of the tubulin superfamily are the constituents of the microtubules and the γ-tubulin meshwork. Mutations in members of the tubulin superfamily are involved in developmental brain disorders, and tubulin activities are the target for various chemotherapies. The intricate functions (game) of tubulins depend on the activities of the GTP-binding domain of α-, β-, and γ-tubulin. This review compares the GTP-binding domains of γ-tubulin, α-tubulin, and β-tubulin and, based on their similarities, recapitulates the known functions and the impact of the γ-tubulin GTP-binding domain in the regulation of the γ-tubulin meshwork and cellular homeostasis.

## 1. Introduction

Proteins known as GTPases hydrolyze GTP to realize a variety of functions in both eukaryotes and prokaryotes. GTPases contain a conserved core GTPase domain with two stable states: one that strongly binds to GTP and the second that is either GDP-bound or GTP/GDP-free. The cycling between GTP and GDP is the essence of the GTPases and forms a cellular switch that orchestrates multiple cellular functions [1]. The GDP-to-GTP transition is accompanied by a conformational change that enables GTPases to transduce signals to interacting proteins, transmitting changes to pathways, such as vesicle trafficking or cellular morphology [2]. Therefore, it is not surprising that GTPases’ dysfunction contributes to the development of multiple diseases [3,4].

The activities of GTPases regulate the dynamics of cytoskeletal elements, such as actin, microtubules, and the γ-tubulin meshwork. The Ras homologous (Rho) family of small GTPase (including Rac, Cdc42, and Rho) modify the actin cytoskeleton, whereas heterodimers of the GTPases α- and β-tubulin (α/β-tubulin) are the constituents of microtubules. The binding of GTP to α-tubulin affects the structure of the protein [5,6]; in contrast, the binding and hydrolysis of GTP to β-tubulin regulate the dynamics of microtubules [7]. Another member of the tubulin superfamily, γ-tubulin, is also a GTPase that binds and hydrolyzes GTP [8,9,10]. γ-Tubulin is the major component of the γ-tubulin meshwork [9,10,11,12,13], but the functions of the GTPase activities of γ-tubulin remain elusive. The possible functions of the GTP-binding domain of γ-tubulin as a regulator of the dynamics of the γ-tubulin meshwork are summarized and discussed in this review based on the latest advances in the field.

## 2. α-, β-, and γ-Tubulin, and Microtubule Formation

Human γ-tubulin is one of the five known tubulin members (γ-, α-, β-, ε-, and δ-tubulin) in the GTPase superfamily of tubulins [14]. A common feature among the members is a conserved GTP-binding domain in the N-terminal region, which consists of five α-helices and six parallel β-strands (Figure 1) [15,16], whereas the C-terminal region is most variable [3].

### 2.1. Microtubule Shrinking and Growing Is GTPase-Dependent

In eukaryotic cells, most of the microtubules comprise 13 laterally associated α/β-tubulin protofilaments, forming hollow fibers in the cytoplasm, axons, and mitotic spindles [17]. The α/β-tubulin heterodimers were first purified as proteins that bind to the depolymerizing natural product colchicine [7]. It was later found that a high concentration of α/β-tubulin can spontaneously nucleate microtubules [18] by the polar head-to-tail association of α/β-tubulin, and required the α/β-tubulin heterodimers bound to GTP [19,20].

In the α/β-tubulin heterodimers, there are two GTP-binding sites: one site on β-tubulin (the E site), and a second site on α-tubulin (the N site) [6]. β-tubulin hydrolyzes GTP during polymerization, and the resulting GDP molecule remains bound, while β-tubulin is part of the polymer. Upon microtubule depolymerization, GDP is exchanged to GTP, and β-tubulin can polymerize once more. In contrast, in α-tubulin, the GTP bound to the N site is neither hydrolyzed nor exchangeable during polymerization [6]. Moreover, the amino acid residues Asp 251 and Glu 254 in α-tubulin stimulate the GTPase activity of β-tubulin (Figure 1a) [21]. The microtubule dynamic is achieved when the GTP molecule in the E site of β-tubulin is hydrolyzed [19,20].

Studies of the structure of αβ-tubulin have revealed that the unpolymerized conformation of α/β-tubulin is curved whether GTP or GDP is bound. Monomeric α- and β-tubulins also have a curved conformation that is reinforced in the heterodimer [22]. As microtubules polymerize and depolymerize, α/β-tubulin can adopt a range of conformations. The transition from curved-to-straight occurs during the polymerization process, and it depends on the binding to GTP [22,23,24], whereas hydrolysis of GTP is necessary for the depolymerization of microtubules [16]. The differences between curved and straight conformations involve rearrangements of the protein region enclosing helices 6 and 7. In addition, the relative orientation between the N-terminal domain and the intermediate domain within tubulin subunits is altered [25]. Mutation of Thr 238 to Ala in β-tubulin creates hyperstable microtubules in yeast, as the mutation impedes the conformational change that the hydrolysis of GTP triggers in β-tubulin when associated with a microtubule [26].

### 2.2. De Novo Nucleation of Microtubules In Vivo Is in Need of Assistance from γ-Tubulin

In vivo, the low concentration of cytosolic α/β-tubulin prevents spontaneous nucleation of microtubules. Thus, in eukaryotes, the nucleation of microtubules occurs from microtubules-organizing centers (MTOCs; centrosomes in animal cells and spindle pole bodies in fungi), which are structures rich in γ-tubulin [27,28]. Microtubule-associated proteins regulate the dynamic behavior of plus and minus ends microtubules [29]. From the minus end, a microtubule grows on a complex formed of γ-tubulin and various γ-tubulin complex proteins (GCPs), the γ-tubulin ring complex (γTURC). A γTURC can both cap the minus ends of noncentrosomal microtubules or anchor microtubules to MTOCs [27,28].

Studies of *Saccharomyces cerevisiae* have shown that only the GTP-TUB4 (ortholog of γ-tubulin) bound to GTP forms a complex that enhanced the interaction of TUB4 with microtubules, facilitating microtubule formation [8], thus implying that both nucleation and elongation of microtubules is regulated by the GTP-binding domain of tubulins. Nonetheless, coexpression of a single-guide RNA (sg) TUBG (CRISPR-Cas9, knocks out TUBG gene that encodes γ-tubulin) and an sgRNA-resistant γ-tubulin GTPase mutant (introduced the expression of a recombinant GTP-binding-defective γ-tubulin) in U2OS cells show no microtubule defects [9]. An explanation for the differences between studies could be the limited reduction in the endogenous γ-tubulin levels in surviving cells transfected with single-guide TUBG [9]. In cell lines, the live imaging of TUBG sgRNA-expressing cells revealed that cells normally divide for several days until a 50% drop in the γ-tubulin protein levels is reached, which leads to apoptotic death [10]. When replacing the endogenous pool of γ-tubulin with an sgRNA-resistant γ-tubulin GTP-binding mutant, the cells survive if sufficient wild-type γ-tubulin is expressed, as the γ-tubulin GTPase mutant cannot suffice for the survival of the cells [9]. The possible presence of small amounts of wild-type γ-tubulin supports microtubule dynamics in this experimental setup [9]. Alternatively, microtubule nucleation may exist in mammalian cells that is independent of the activities of the GTP-binding domain of γ-tubulin.

## 3. The GTP-Binding Domain of γ-Tubulin

The homology between the protein sequences of the GTPase domains of α-tubulin (residues 1–254 in Figure 1a) and β-tubulin (residues 1–252 in Figure 1a) is 63%. In contrast, the protein sequence in the GTPase domain of γ-tubulin (residues 1–255 in Figure 1a) exhibits 60%, and 64%, homology with the corresponding sequences in α-tubulin and β-tubulin, respectively, suggesting that the nucleotide-binding site in the GTPase domain of γ-tubulin is most similar to the E site (β-tubulin) rather than to the N site (α-tubulin).

The crystal structure of γ-tubulin is similar to the structures of α- and β-tubulin (Figure 1b) [16,25,31]. Compared with the N terminal region of α- and β-tubulin, γ-tubulin has one deletion and three insertions (Figure 1a). γ-Tubulin lacks a His residue between residues 107 and 108; this probably alters the α-helix structure formed by the amino acid sequence in this region [25]. The insertions of His-Gly and Asp residues at position 96 and 97 and at position 176, respectively, are part of disorder regions (with not define secondary structure) in the γ-tubulin structure and might be involved in longitudinal contact of the surface of γ-tubulin with other proteins [25].

### 3.1. γ-Tubulin Is an Important Constituent of γ-Tubules and γ-Strings

γ-Tubulin is a ubiquitously expressed protein that was described as being associated with all the cytosolic compartments as well as with the nucleus [32]. Cellular γ-tubulin is the core component of a variety of structures. Due to its self-polymerizing features, the type II chaperonin-containing TCP1 (CCT) fold γ-tubulin into γ-tubulin threads called γstrings [33]. The assembly of γ-strings with recombinant γ-tubulin occurs in vitro in the absence of GTP (Figure 2a) and assists in the formation of lamin B3 protofilaments [12]. Within eukaryotic cells, cytosolic and nuclear γ-strings (Figure 2b) associate with centrosomes and cellular membranes and assist in the formation of the nuclear compartment around chromatin [10,12,34]. A meshwork of mitochondrial-DNA-associated γ-strings provides mitochondria with a structural scaffold that regulates the function of those organelles [10]. Other γ-tubulin structures are γ-tubules, which are larger γ-tubulin filaments that nucleate on and are formed of cytosolic foci of γTURCs together with the centrosomal protein pericentrin (Figure 2b) [9,35]. γ-Tubules are protein structures in cells that are affected by a variety of treatments, such as cold, Colcemid, and Taxol [9].

### 3.2. The Function of the GTP-binding Domain of γ-Tubulin

The residues Cys^12^, Asn^226^, Asn^204^, Gln^11^, Gly^142^, Gly^144^, and Thr^143^ in β-tubulin interact with GTP and are conserved in γ-tubulin (Figure 1). Studies of the crystal and three-dimensional structure of the GTP-binding domain of human γ-tubulin predict that the ribose moiety of GTP interacts with Ser^139^ and Asn^207^, whereas the guanine base interacts with Asn^207^, Phe^225^, and Asn^229^. The phosphates in GTP contact Gln^12^, Cys^13^, Ser^139^, and Thr^145^, whereas the ion Mg^2+^ interacts with Asp^68^ and Glu^70^ (Figure 1) [8,25]. Analysis of human U2OS osteosarcoma cells and *S. cerevisiae* revealed that mutations in the GTP-interacting residue Cys^13^ of γ-tubulin are cytotoxic [8,9,10,25]. Moreover, compounds that interfere with the GTP binding in the GTPase domain of γ-tubulin have a cytotoxic effect in tumors lacking a functional retinoblastoma pathway [30,36,37]. Earlier work demonstrated that the acetal moiety of the citral analog, citral dimethyl acetal (CDA), binds to the GTP-binding domain of γ-tubulin similarly to the guanine base of GTP by interacting with Cys^13^, Asn^207^, Phe^225^, and Asn^229^ (Figure 1c). After the hydrolysis of the dimethyl acetal to an aldehyde, the modified compound binds covalently to Cys^13^, interfering with the GTPase activity of γ-tubulin [30]. Another compound that also interferes with the functions of Cys^13^ in γ-tubulin is dimethyl fumarate (DMF), an approved drug for the treatment of multiple sclerosis and psoriasis [30]. DMF is a cell-permeable derivative of the metabolite fumarate, which is endogenously produced under oxidative stress [9,10,30,38]. Thus, the cellular levels of fumarate may locally provide a regulatory link between cell metabolism and the dynamics of the γ-tubulin meshwork. Treatment with CDA, DMF, or stable expression of the Cys^13^-to-Ala-γ-tubulin mutant disassembles γ-tubules and affects the respiratory capacity of cells [9,10,11,30]. Therefore, it was proposed that GTP may control the dynamics of the γ-tubulin meshwork.

### 3.3. The Conformational Switch of γ-Tubulin

The structural similarity between the nucleotide-binding pockets of γ- and β-tubulin results in identical nucleotide-binding affinities [25], which strongly suggest that γ- and β-tubulin share common nucleotide-binding energetics that are largely unaffected by the above-mentioned differences. α-, β-, and γ-tubulin share many of the features responsible for GTP binding [16,31,39]. Based on those findings and the interaction between α-, and γ-tubulin, it was suggested that the longitudinal association of α-tubulin might also stimulate the GTPase activity of γ-tubulin [25]. Accordingly, mutational studies in *S. cerevisiae* [8] confirmed that interactions between α-tubulin and γ-tubulin might induce GTP hydrolysis in γ-tubulin [8], and a model was proposed in which GTP binding to γ-tubulin regulates interactions with α-tubulin, which stimulate the GTPase activity of γ-tubulin in a similar way to the function of GTP-bound β-tubulin [8]. Nonetheless, as α-tubulin is not found in all cellular compartments, there must be additional factors that regulate the GTPase activities of γ-tubulin.

Despite the mentioned similarities between α-, β-, and γ-tubulin, studies of the crystal structure of γ-tubulin–GTPγS have shown a curved conformation that resembles that observed for unpolymerized α-β-tubulin–GDP [25], leading to the hypothesis that the hydrolysis of GTP is not a conformational switch in γ-tubulin. Still, similar to Thr 238 in β-tubulin [26], Thr 241 is conserved in γ-tubulin, and mutations of Thr 241-Thr 242 to Ala-Ala-γ-tubulin also reduced the shrinkage rate of γ-tubules in U2OS cells [9], suggesting that γ-tubulin undergoes conformational switches and that those conformational states might be assisted by the hydrolysis of GTP in combination with other biochemical modifications or stimuli.

#### Phosphorylation of γ-Tubulin

By changing its location during cell division, γ-tubulin is an important regulator of both centrosome dynamics and of S-phase progression [37,40,41,42,43]. Studies in mammalian cell lines and proteomic analysis of mitotic centrosomes established that, at the centrosome, γ-tubulin is phosphorylated (P) on Ser^131^ [44,45]. High levels of Ser(P)^131^-γ-tubulin reduce astral microtubule nucleation at the centrosomes, enhance γ-tubulin polymerization at the nascent centriole, and prevent the acentriolar formation of centrosomes [44,45]. Ser(P)^131^-γ-tubulin in the centrosome also facilitates the accessibility to Ser^385^. Ser (P)^385^-γ-tubulin then triggers the release of γ-tubulin from microtubules and starts the accumulation of γ-tubulin in the nuclear compartment, resulting in the moderation of gene expression [36,37,42].

Overall, it is plausible that phosphorylations of Ser^131^- and Ser^385^-γ-tubulin induce different conformational states. Both residues are part of disorder regions in the γ-tubulin structure. The three-dimensional structure of γ-tubulin places Ser^131^ between the γ-tubulin lateral interaction surface and the γ-tubulin region that controls GTP binding [25,46]; in contrast, Ser^385^ is placed close to the nuclear localization signal of γ-tubulin at the start of a helix–loop–helix motif [3,42]. A helix–loop–helix is a commonly found motif in DNA-binding proteins, which concurs with the DNA-binding ability of γ-tubulin [11,37]. Therefore, Ser(P)^131^ may stabilize γ-tubulin in a polymerized conformation [44], whereas Ser (P)^385^ may assist in the exposure of the C-terminal nuclear localization signal on γ-tubulin [37,42].

Several empirical observations support the existence of several conformational states. First, the anti-Ser (P)^131^-γ-tubulin antibody labeled only one centriole, and electron microscopy analysis revealed the presence of the Asp^131^-γ-tubulin phosphomimetic mutant on the surface of the centrosome [44]. Second, expression of Asp^131^-γ-tubulin increased centriole duplication but reduced astral microtubule regrowth [44]. Third, Ser (P)^385^-γ-tubulin undergoes a size shift in SDS gels [42]. Fourth, neither anti-total γ-tubulin nor -GFP antibodies recognize the chromatin-bound phosphomimetic Ser^385^ to Asp γ-tubulin mutants in immunofluorescence analysis [42]. Finally, immunofluorescence analysis with an anti-total γ-tubulin showed that the antibodies recognize only part of the chromatin-associated pool [12]. Together, the data strongly support the existence of several conformational states in γ-tubulin.

## 4. The Dynamics of the γ-Tubulin Meshwork

There are many characteristics that are shared among the members of the tubulin superfamily; together, the members are important proteins in the development of the brain as well as in the fight against cancer [47,48,49,50,51,52]. The structural plasticity and dynamics of microtubules depend on the activities of the GTP-binding domain of α-, β-, and γ-tubulin [8,53]. Similarly, one should expect that the GTP-binding domain in tubulins also regulates the structural plasticity and dynamics of the γ-tubulin meshwork.

Both γ-tubulin and its GTP-binding domain are necessary for cell survival [8,9,10]. Though, it is unclear how the GTPase contributes to the function of the protein. The canonical functions of γ-tubulin start in the S phase with the duplication of centrosomes that, at the onset of mitosis, ensures the assembly of a bipolar mitotic spindle and the segregation of sister chromatids between offspring cells. γTURCs also nucleate cytosolic microtubules in the interphase. Apart from these well-studied functions, γ-tubulin forms a meshwork. However, we know too little about how the meshwork is regulated, its functions, and its dynamics. The problems in elucidating the functions of γ-tubulin are that many of the cell structures containing γ-tubulin are not preserved after preparation of the samples [35], and available antibodies do not recognize the whole cellular pool of γ-tubulin [12,42]. In addition, a cell model with a total CRISPR-Cas9-mediated knockout of γ-tubulin is impossible to achieve as γ-tubulin is essential for cell survival [10]. In contrast, short hairpin (sh)RNA-mediated reduction of γ-tubulin expression results in stable cell lines that express half the amount of the total γ-tubulin pool [36], providing a useful tool for studying the functions of the γ-tubulin meshwork [10,36].

Thus, to increase our knowledge of the meshwork, live imaging of cells with a fluorescence label meshwork may provide us with hints on the biological functions of γ-tubulin and its GTPase. In time-lapse images of living U2OS cells that stably express both TUBG-shRNA (which lowers the endogenous pool of γ-tubulin by ~50% [36]) and a C-tagged TUBG1-green fluorescence protein (GFP) shRNA-resistant gene (which fluorescence labels the γ-tubulin meshwork [10,43]), it is visible that, during the interphase, γ-strings are found in the cytoplasm and nucleus, whereas centrosomes and γ-tubules are in the cytosol and their positions are constantly changing (Figure 3 and Appendix A) [11,43]. It is apparent that γ-tubules can be formed close to the nuclear envelope and that centrosomes can either nucleate γ-tubules or move along and among them (Figure 3 and Appendix A). The location of the centrosomes varies from cell to cell. Centrosomes are located on the cytosolic side of the nuclear envelope or in the cytosol, but in both cases, their position is constantly changing (Figure 3 and Appendix A). The number, location, and length of γ-tubules also vary in both cells and overtime (Figure 3 and Appendix A).

γ-Strings span from the cytosolic compartment through the nuclear membrane into the chromatin and can emanate from centrosomes and γ-tubulin rich structures (Figure 3 and Appendix A). These foci localize to both the cytoplasm and nucleus and γ-strings can emanate from those structures (Figure 3 and Appendix A). Notably, the nuclear foci are located towards the interior, where genes normally are positioned (Figure 3 and Appendix A) [32]. An earlier study using various mammalian cell lines demonstrated that γ-tubules nucleate on cytosolic aggregates or γTURC foci together with pericentrin [9]. Both γTURCs and pericentrin are cytosolic proteins, which strongly suggests that the contents of the nuclear and cytosolic foci differ, but the common component is γ-tubulin [9].

## 5. Conclusions and Future Perspectives

The present review aimed to summarize the current knowledge of the function of the GTP-binding domain of γ-tubulin in cellular homeostasis and speculate on the possible roles of the GTP-binding domain in the organization of the γ-tubulin meshwork. Still, further insights are needed to elucidate the impact of the GTP-binding domain of γ-tubulin in health and disease.

In my opinion, there are three unique features in the γ-tubulin meshwork. First, the meshwork joins together the cytoplasm with the mitochondrial and nuclear genome [10,12,37]. Second, the meshwork is described to interact with signal pathways known to be essential for cell survival and disease development [3,11,32,43]. Third, the presence of γ-tubulin in cellular membranes and cytoskeletal elements, as well as in organelles, provides a meshwork with plasticity.

In eukaryotes, the creation of membrane-enclosed compartments is the prerequisite for transport, communication, and energy production [54]. A variety of cytoskeletal networks further organized this complex system of membranes [3]. For example, the GTP-binding domain of γ-tubulin is required for normal mitochondrial function and spatial organization [10]. Most likely, the presence of γ-tubulin in membranes, together with the sticky nature of the protein and the activities of its GTP-binding domain, provides a cellular environment that facilitates the spatial organization of membranes. This is achieved by the interaction of membrane-associated γ-tubulin with other components of the cytosolic/nuclear γ-tubulin meshwork, which, in turn, links the membranes to other cytoskeleton/nucleoskeleton platforms, such as microtubules, nuclear lamins, and actin filaments [12,55,56,57].

Based on live imaging, it is tempting to speculate on how the GTP-binding domain regulates the meshwork. The basis of the meshwork is the folding of the newly translated cytosolic γ-tubulin into γ-strings independently of γ-tubulin’s GTPase by the chaperonin CCT [12,33]. Part of these protofilaments can then arrange in a GTP-dependent manner into γ-tubules [9]. The transport of γ-tubulin into the nuclear compartment and the association of the protein with the centrosomes is most likely GTP dependent [42,44]. The association of γ-strings with membranes is probably initiated during the formation of the nuclear envelope around the mitotic chromatin-associated γ-tubulin boundary [12]. The formation of γ-tubulin foci is most likely GTP-dependent and may function as a nucleating scaffold for γ-strings and γ-tubules. However, at present, we do not understand the mechanism behind the constantly changing γ-tubulin meshwork. An intriguing question is if the constant movement of centrosomes and γ-tubules and the rapid formation and dissolution of γ-tubulin foci are dependent on the GTPase activities of γ-tubulin.

## Figures and Tables

**Figure 1 cells-10-00745-f001:**
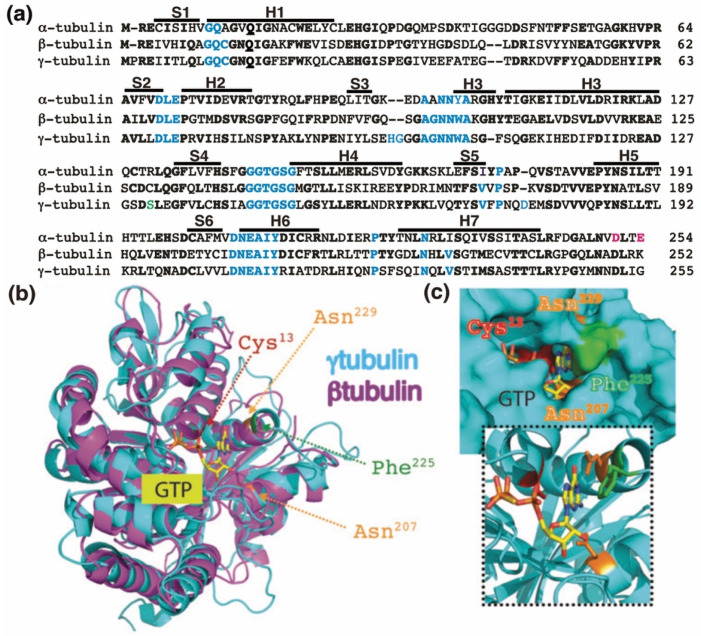
The GTP-binding domain. (**a**) Sequence alignment of the N-terminal GTP-binding domain (enclosing the first five α-helices and six parallel β-strands [16]) and nearby regulating sequences of human α-tubulin (residues 1–254; GenBank: AAA91576), β-tubulin (residues 1–252; GenBank: AAB59507.1), and γ-tubulin (residues 1–255; GenBank: AAV38734.1). Bold and magenta letters indicate identical and essential residues for the β-tubulin-mediated hydrolysis of GTP, respectively [21]. Residues involved in GTP binding are blue. Ser^131^ is labeled green. Helices (H) and strands (S) are indicated. (**b**) The known three-dimensional structures of β-tubulin (magenta) and γ-tubulin (cyan) with a view down the GTP-binding pocket, highlighting Cys^13^, Asn^207^, Phe^225^, and Asn^229^. (**c**) The molecular surface of the GTP/GDP-binding pocket of γ-tubulin is displayed, and GTP is shown in a stick representation. The inset (dashed lines) shows GTP together with the protein with only the side-chains of Cys^13^, Asn^207^, Phe^225^ and Asn^229^ displayed. (**b**,**c**) Reproduced with permission from B.O. Villoutreix; published by American Association for Cancer Research, 2015 [30].

**Figure 2 cells-10-00745-f002:**
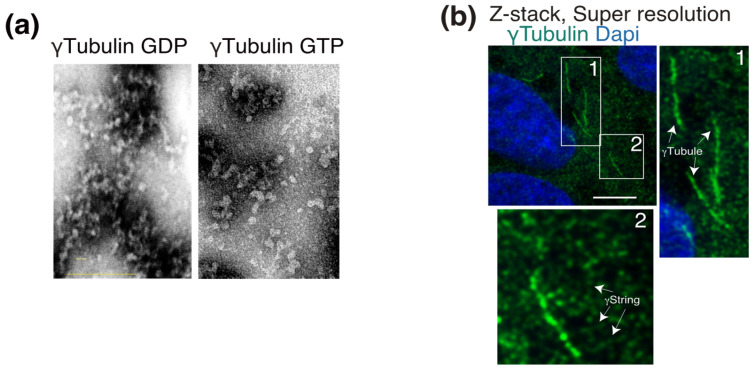
γ-Tubulin is the main constituent of γstrings and γ-tubules. (**a**) Purified γ-tubulin was negatively stained and imaged using electron microscopy in the absence or presence of 1 mM GTP, as indicated. (**b**) Endogenous γ-tubulin was detected with an anti-γ-tubulin antibody, and nuclei were detected with DAPI in fixed U2OS cells. Airyscan super-resolution microscopy was performed to capture the faintest signals emitted by thin γstrings and reveal the γ-tubulin meshwork details. Z-stack shows average intensity projection of 14 sequential Airyscan super-resolution images that were collected at 0.2-μm intervals. The white boxes indicate the magnified area shown in the insets. White arrows indicate γ-tubules and γstrings, as indicated. Scale bars: 10 μm. (**a**,**b**) These data are from unpublished works by Alvarado-Kristensson et al.

**Figure 3 cells-10-00745-f003:**
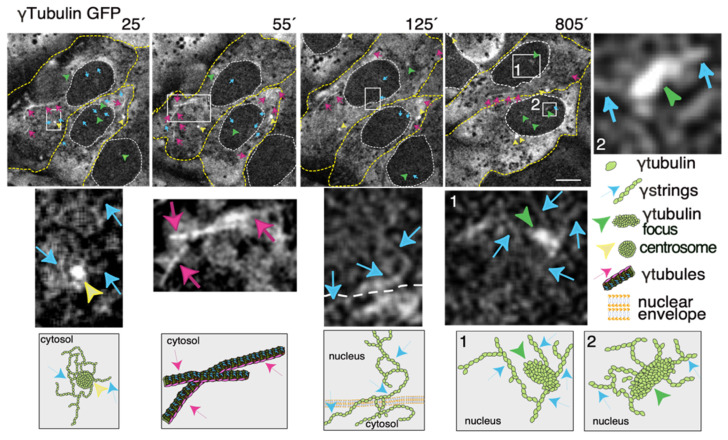
The dynamics of the γ-tubulin meshwork in living cells. Time-lapse series of fluorescence images showing U2OS cells stably coexpressing TUBG-short hairpin (sh)RNA and GFP-tagged sh-resistant γ-tubulin (γtubulin GFP). Images present chosen frames illustrating the changes in the position of γ-strings, γ-tubules, centrosomes, and γ-tubulin foci during the interphase. The outer membrane of the cells and the nuclei are indicated by dotted yellow and white lines, respectively. Yellow arrowheads, blue arrows, and magenta arrows indicate centrosomes, γstrings, and γ-tubules, respectively. Green arrowheads denote nucleating γ-tubulin foci. The images shown were collected every 5 min and represent selected frames showing the changes that γstrings, γ-tubules, centrosomes, and γ-tubulin foci undergo during interphase. White boxes indicate the areas magnified in the inset. Scale bars: 10 μm. The lower panels are schematic representations of the structures shown in the corresponding white box. See Appendix A. The stable cell line was obtained, and time-lapse experiments were performed as previously described [10,12,35]. These data are from unpublished works by Alvarado-Kristensson et al.

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
