# Peer review of "The Game of Tubulins"

_cells, 2021, doi:10.3390/cells10040745_

Round 1

Reviewer 1 Report

The review by Maria Alvarado Kristensson is describing microtubule structure, biochemistry, and formation with a special focus on g tubulin. This is a very interesting topic that fits well to the scope of Cells. However, in the current form the review is often difficult to understand for researchers not working with tubulins. More explanatory sentences and also more figures illustrating organization and biological function of tubulins are required to increase the readability.

More specific comments:

  1. 2: Meaning of the title unclear

  1. 7: The abstract is currently only a long headline. Maybe one can include some conclusions with respect to the biology?

  1. 73: De Novo nucleation?

  1. 83: This paragraph is very important, but hard to understand for those that have not read the original studies. It should be rewritten and maybe expanded to facilitate the understanding of reported research.
  2. 83: TUBG1 was not introduced before. Do yeast orthologues exists to all human microtubule? Are there yeast specific microtubule? Little extra information on the similarities and differences between yeast and human will be helpful for the reader to interpret the yeast data better.
  3. 86: single guide RNA
  4. 86: The experiment is not clearly described. The endogenous g tubulin was deleted and non cuttable, but otherwise normal g tubulin was ectopically expressed? Why should a phenotype be expected if the ko is rescued?
  5. 89: It is unclear what is meant by the “limited reduction” as a CRISPR induced ko will lead to a 100% reduction per ko cell. Cell lethal ko should lead to complete loss of cells.

  1. 98: “In this context” is not clear. How is the GTPase domain homology related to what was described in the previous paragraph? The sequence comparision might fit better to the next paragraph “The GTPase Domain of g-tubulin”.

  1. 139: It would be much easier to follow this part if one would see a 3D structure of tubulin. Marking of GTP binding residues by colour instead of just underlining would also be helpful. There should be many more numbers in Fig. 1 in order to help the reader to find the aa mentioned.
  2. 140: According to fig. 1 Cys is aa 13 not 12.
  3. 141: “Other important conserved 141 residues are: Gln11, Gly142, Gly144, and Thr143 (Figure 1)” Why are they important?
  4. 142: “Gln11” According to Fig. 1 there is a G (glycin) at aa 11 and a Q (Gln) at aa 12.
  5. 145: The phosphates are not contacted by hydrogen bonds? (cf. L. 140).

  1. 169: “Based on those findings, it was suggested that the longitudinal association of α-tubulin may also stimulate the GTPase activity of g-tubulin [28]” This is not evident from the findings mentioned before. More explanation needed. (Association of a-tubulin with g-tubulin?)

  1. 172: “model was proposed in 172 which GTP binding to g -tubulin regulates interactions with tubulins in a similar way to 173 the function of GTP-bound β-tubulin [8]”. Again, more information is needed to understand this sentence. Binding of g tubulin to a, b, and g tubulin? The function of GTP bound b-tubulin in relation to what?

  1. 182: “Additionally, several lines of evidence suggest that-tubulin undergo conformational switches and that those conformational states might be assisted by the hydrolysis of GTP in combination with other biochemical modifications or stimuli.” These lines of evidence should be discussed, referenced, and the author should state her own opinion in this matter.

  1. 189: “proteonomic analysis” proteomic?

  1. 198: It should be defined what is meant by “disorder regions”

  1. 244: It would be nice to discuss the consequences of shRNA and sgRNA on TUBG together, if similar parameter such as survival had been tested.

  1. 268: It is unclear how “nevertheless” is related to the previous sentence which presented the aim of the review.

  1. 277: What is known about the g tubulin meshwork should have been already mentioned L 224, where the meshwork is introduced.

Author Response

More specific comments:

  1. 2: Meaning of the title unclear

 I have explained better the meaning of the title in the abstract. The game represents the in intricate relation between the functions of tubulins.

  1. 7: The abstract is currently only a long headline. Maybe one can include some conclusions with respect to the biology?
  1. 73: De Novo nucleation?
  1. 83: This paragraph is very important, but hard to understand for those that have not read the original studies. It should be rewritten and maybe expanded to facilitate the understanding of reported research.
  2. 83: TUBG1 was not introduced before. Do yeast orthologues exists to all human microtubule? Are there yeast specific microtubule? Little extra information on the similarities and differences between yeast and human will be helpful for the reader to interpret the yeast data better.
  3. 86: single guide RNA
  4. 86: The experiment is not clearly described. The endogenous g tubulin was deleted and non cuttable, but otherwise normal g tubulin was ectopically expressed? Why should a phenotype be expected if the ko is rescued?

  I modified the text according to your suggestions.

  1. 89: It is unclear what is meant by the “limited reduction” as a CRISPR induced ko will lead to a 100% reduction per ko cell. Cell lethal ko should lead to complete loss of cells.

 I modified the text according to your suggestions.

  1. 98: “In this context” is not clear. How is the GTPase domain homology related to what was described in the previous paragraph? The sequence comparision might fit better to the next paragraph “The GTPase Domain of g-tubulin”.

 I have moved the paragraph to the beginning of section 3.

  1. 139: It would be much easier to follow this part if one would see a 3D structure of tubulin. Marking of GTP binding residues by colour instead of just underlining would also be helpful. There should be many more numbers in Fig. 1 in order to help the reader to find the aa mentioned.

A 3D structure has been added to Figure 1b.

  1. 140: According to fig. 1 Cys is aa 13 not 12.

This is referring to beta-tubulin. The sequence is as follow: MREIVHIQAGQC 12

  1. 141: “Other important conserved 141 residues are: Gln11, Gly142, Gly144, and Thr143 (Figure 1)” Why are they important?
  2. 142: “Gln11” According to Fig. 1 there is a G (glycin) at aa 11 and a Q (Gln) at aa 12.
  3. 145: The phosphates are not contacted by hydrogen bonds? (cf. L. 140).
  4. 169: “Based on those findings, it was suggested that the longitudinal association of α-tubulin may also stimulate the GTPase activity of g-tubulin [28]” This is not evident from the findings mentioned before. More explanation needed. (Association of a-tubulin with g-tubulin?)
  5. 172: “model was proposed in 172 which GTP binding to g -tubulin regulates interactions with tubulins in a similar way to 173 the function of GTP-bound β-tubulin [8]”. Again, more information is needed to understand this sentence. Binding of g tubulin to a, b, and g tubulin? The function of GTP bound b-tubulin in relation to what?

I have modified the text so it is easier to understand. Regarding glycin, this is referring to beta-tubulin. The sequence is as follow: MREIVHIQAGQ 11

  1. 182: “Additionally, several lines of evidence suggest that-tubulin undergo conformational switches and that those conformational states might be assisted by the hydrolysis of GTP in combination with other biochemical modifications or stimuli.” These lines of evidence should be discussed, referenced, and the author should state her own opinion in this matter.

 The text has been modified.

  1. 189: “proteonomic analysis” proteomic?

 Thank you for your comment. The error has been corrected in the revised version of the manuscript.

  1. 198: It should be defined what is meant by “disorder regions”

 This is explained in the revised version of the manuscript

  1. 244: It would be nice to discuss the consequences of shRNA and sgRNA on TUBG together, if similar parameter such as survival had been tested.

 I have added an explanation to the differences between shRNA and sgRNA on TUBG in the revised version of the manuscript.

  1. 268: It is unclear how “nevertheless” is related to the previous sentence which presented the aim of the review.

 I have removed nevertheless.

  1. 277: What is known about the g tubulin meshwork should have been already mentioned L 224, where the meshwork is introduced.

 I have moved the texted as suggested.

Thank you very much for your helpful comments, time and consideration.

Reviewer 2 Report

In this review article, Alvarado Kristensson compares the GTP binding domain of alpha-, beta- and gamma-tubulin and presents the current data on the regulation of gamma-tubulin function by GTP hydrolysis. Overall, I find that the review is mainly descriptive, lacking a critical vision, and is written without a true focus. An extensive rewriting is needed.

Some (more specific) comments are given below.

1) The use of “tubulin isoforms” is confusing (line 37). This term is usually used to define the diversity (isotypes and post-translational modifications) within a tubulin family member. Why not use “members” as in the cited Dutcher’s review? In addition, Dutcher listed 7 members of the tubulin family, including eta- and epsilon-tubulin. Is there any reason these last tubulins are not mentioned in this review?

2) I also find the use of “GTPase domain” misleading. Indeed, the GTP molecule bound to alpha-tubulin is not hydrolyzed, and the hydrolysis of the one bound to beta-tubulin is catalyzed by alpha-tubulin residue of another heterodimer through longitudinal tubulin-tubulin association. “GTP-binding domain” or “nucleotide binding domain” would be more appropriate.

3) A better structural description of the tubulin GTP-binding domain is absolutely required. In particular (but not limited to these points):

- the author writes that this domain consists of five alpha-helices, while Figure 1 shows 7 alpha-helices, and 6 beta-sheets. I surmise that there is one beta-sheet of 6 beta-strands?

- Why only the sequences of alpha-, beta, and gamma-tubulin are shown in Fig. 1, and not those of the other human tubulin members? In addition, the sequence conservation (using, e.g., clustal style) could be indicated.

- It seems that the secondary structural elements in Fig. 1 are those originally defined by Nogales and colleagues (Nature, 1998). Since then, higher resolution structures have become available and the definition and boundaries of the secondary structural elements have been refined.

- Whereas the review deals with the nucleotide-binding domain of tubulin, there is no figure of structures. Several are needed to put in a structural context the many residues that are mentioned, to illustrate the nucleotide binding pocket, the “curved” conformation of tubulin monomers (the author should define what a “curved conformation” of monomeric tubulin exactly means), …

4) The sentence lines 45-48 suggests that GTP hydrolysis in beta-tubulin is required for microtubule nucleation, which is actually the reverse: GTP hydrolysis inhibits microtubule nucleation (Carlier et al, Biophys J, 1997).

5) Lines 59-60: “The hydrolysis of GTP is necessary for the switching of the heterodimer between two conformations”. The situation is more complex than that, since tubulin either bound to GDP or to a stable GTP analog is straight in the microtubule core and curved in the available crystal structures.

Author Response

  1. The use of “tubulin isoforms” is confusing (line 37). This term is usually used to define the diversity (isotypes and post-translational modifications) within a tubulin family member. Why not use “members” as in the cited Dutcher’s review? In addition, Dutcher listed 7 members of the tubulin family, including eta- and epsilon-tubulin. Is there any reason these last tubulins are not mentioned in this review?

 I changed isoforms to members, as suggested. Eta and epsilon are not found in humans. The focus of the whole review is the functions of the members of the tubulin family in humans.

  1. I also find the use of “GTPase domain” misleading. Indeed, the GTP molecule bound to alpha-tubulin is not hydrolyzed, and the hydrolysis of the one bound to beta-tubulin is catalyzed by alpha-tubulin residue of another heterodimer through longitudinal tubulin-tubulin association. “GTP-binding domain” or “nucleotide binding domain” would be more appropriate.

Thank you for your comment, and yes, I agree that in the way you presented the term GTPase domain is misleading. I have replaced it in the text of the revised version of the manuscript to GTP-binding domain.

3) A better structural description of the tubulin GTP-binding domain is absolutely required. In particular (but not limited to these points):

- the author writes that this domain consists of five alpha-helices, while Figure 1 shows 7 alpha-helices, and 6 beta-sheets. I surmise that there is one beta-sheet of 6 beta-strands?

Thank you for your comment. We have clarified in the figure 1 legend that the additional helices enclose nearby regulating sequences.

- Why only the sequences of alpha-, beta, and gamma-tubulin are shown in Fig. 1, and not those of the other human tubulin members? In addition, the sequence conservation (using, e.g., clustal style) could be indicated.

The review focus on the major components of microtubules and the gamma-tubulin meshwork in humans. Thus, we only discuss alpha-, beta-, and gamma-tubulin.

- It seems that the secondary structural elements in Fig. 1 are those originally defined by Nogales and colleagues (Nature, 1998). Since then, higher resolution structures have become available and the definition and boundaries of the secondary structural elements have been refined.

Thank you for your comment, we have updated figure 1a.

- Whereas the review deals with the nucleotide-binding domain of tubulin, there is no figure of structures. Several are needed to put in a structural context the many residues that are mentioned, to illustrate the nucleotide binding pocket, the “curved” conformation of tubulin monomers (the author should define what a “curved conformation” of monomeric tubulin exactly means), …

In the current version of the manuscript, we have added to figure 1b the three-dimensional structure alignment of beta- and gamma-tubulin bound to GTP.

  • The sentence lines 45-48 suggests that GTP hydrolysis in beta-tubulin is required for microtubule nucleation, which is actually the reverse: GTP hydrolysis inhibits microtubule nucleation (Carlier et al, Biophys J, 1997).

Thank you for your comment, we have now mended the mistake.

5) Lines 59-60: “The hydrolysis of GTP is necessary for the switching of the heterodimer between two conformations”. The situation is more complex than that, since tubulin either bound to GDP or to a stable GTP analog is straight in the microtubule core and curved in the available crystal structures.

Thank you for your comment, we have now rewritten the text according to the mentioned research.

Thank you for your helpful comments, time and consideration.

Round 2

Reviewer 1 Report

All comments were addressed sufficiently!

Author Response

Once more, thank you for your time and consideration.

Reviewer 2 Report

In the revised version of the “The Game of Tubulins” review manuscript, Alvarado Kristensson has made cosmetic modifications but the main criticisms I had on the previous version are still relevant.

In particular, several fundamental points that I raised have not been addressed. New Fig. 1B is welcome and shows that gamma-tubulin and beta-tubulin share a similar structure. But the function in the binding of the nucleotide of the many cited residues would benefit from additional figures (close-up of the nucleotide binding site).

In addition, the author speaks about “curved” and “straight” conformations of monomeric tubulin. The curved and straight conformations of alpha-beta-tubulin primarily refer to the orientation of one subunit with respect to the other one. The author should define what is a “curved/straight” conformation of monomeric tubulin.

Minor comments:

- There is a confusion between beta-strands and beta-sheets (https://en.wikipedia.org/wiki/Beta_sheet).

- A proofreading a needed. For instance:

- Lines 33-34: “GTPase that binds and hydrolases GTP” > “…hydrolyzes GTP”?

- Beginning of page 2: "most of the microtubules comprise 13 laterally associated … protofilaments that associate laterally”. One “laterally associated/associate laterally” is enough.

Author Response

In the revised version of the “The Game of Tubulins” review manuscript, Alvarado Kristensson has made cosmetic modifications but the main criticisms I had on the previous version are still relevant.

In particular, several fundamental points that I raised have not been addressed. New Fig. 1B is welcome and shows that gamma-tubulin and beta-tubulin share a similar structure. But the function in the binding of the nucleotide of the many cited residues would benefit from additional figures (close-up of the nucleotide binding site).

In the revised version of the manuscript, I have included a cartoon showing the GTP-binding domain of gamma-tubulin bound to GTP (Figure 1c).

In addition, the author speaks about “curved” and “straight” conformations of monomeric tubulin. The curved and straight conformations of alpha-beta-tubulin primarily refer to the orientation of one subunit with respect to the other one. The author should define what is a “curved/straight” conformation of monomeric tubulin.

The differences between curved and straight conformations are now included in lines 67 to 70.

Minor comments:

- There is a confusion between beta-strands and beta-sheets (https://en.wikipedia.org/wiki/Beta_sheet).

Thank you for your comment. I have mended the mistakes.

- A proofreading a needed. For instance:

- Lines 33-34: “GTPase that binds and hydrolases GTP” > “…hydrolyzes GTP”?

- Beginning of page 2: "most of the microtubules comprise 13 laterally associated … protofilaments that associate laterally”. One “laterally associated/associate laterally” is enough.

Thank you for your comment. I have mended the mistakes and an English-speaking college has read the manuscript.

Once more, thank you for your helpful comments, time and consideration.

Round 3

Reviewer 2 Report

The manuscript is improved and I have no major remaining question.